# Effective Remediation of Arsenic-Contaminated Soils by EK-PRB of Fe/Mn/C-LDH: Performance, Characteristics, and Mechanism

**DOI:** 10.3390/ijerph19074389

**Published:** 2022-04-06

**Authors:** Zongqiang Zhu, Shuai Zhou, Xiaobin Zhou, Shengpeng Mo, Yinian Zhu, Lihao Zhang, Shen Tang, Zhanqiang Fang, Yinming Fan

**Affiliations:** 1Collaborative Innovation Center for Water Pollution Control and Water Safety in Karst Area, Guilin University of Technology, Guilin 541004, China; zhuzongqiang@glut.edu.cn (Z.Z.); zhoushuai202204@163.com (S.Z.); moshengpeng14@mails.ucas.ac.cn (S.M.); zhuyinian@glut.edu.cn (Y.Z.); lhzhang@glut.edu.cn (L.Z.); tangshen@glut.edu.cn (S.T.); 2The Guangxi Key Laboratory of Theory and Technology for Environmental Pollution Control, Guilin University of Technology, Guilin 541004, China; 3Technical Innovation Center of Mine Geological Environmental Restoration Engineering in Southern Karst Area, Nanning 530022, China; 4School of Chemistry and Environment, South China Normal University, Guangzhou 510006, China; zhqfang@scnu.edu.cn

**Keywords:** electrokinetics, permeable reactive barrier, arsenic-contaminated soil, influence factor, remediation mechanism

## Abstract

Arsenic is highly toxic and carcinogenic. The aim of the present work is to develop a good remediation technique for arsenic-contaminated soils. Here, a novel remediation technique by coupling electrokinetics (EK) with the permeable reactive barriers (PRB) of Fe/Mn/C-LDH composite was applied for the remediation of arsenic-contaminated soils. The influences of electric field strength, PRB position, moisture content and PRB filler type on the removal rate of arsenic from the contaminated soils were studied. The Fe/Mn/C-LDH filler synthesized by using bamboo as a template retained the porous characteristics of the original bamboo, and the mass percentage of Fe and Mn elements was 37.85%. The setting of PRB of Fe/Mn/C-LDH placed in the middle was a feasible option, with the maximum and average soil leaching toxicity removal rates of 95.71% and 88.03%, respectively. When the electric field strength was 2 V/cm, both the arsenic removal rate and economic aspects were optimal. The maximum and average soil leaching toxicity removal rates were similar to 98.40% and 84.49% of 3 V/cm, respectively. Besides, the soil moisture content had negligible effect on the removal of arsenic but slight effect on leaching toxicity. The best leaching toxicity removal rate was achieved when the soil moisture content was 35%, neither higher nor lower moisture content in the range of 25–45% was conducive to the improvement of leaching toxicity removal rate. The results showed that the EK-PRB technique could effectively remove arsenic from the contaminated soils. Characterizations of Fe/Mn/C-LDH indicated that the electrostatic adsorption, ion exchange, and surface functional group complexation were the primary ways to remove arsenic.

## 1. Introduction

Arsenic has attracted broad concerns due to its high toxicity and strong carcinogenicity to human beings and animals [1,2]. Traditionally, the remediation technologies of arsenic-contaminated soil include solidification/stabilization, phytoremediation, soil leaching, and electrokinetics (EK) remediation [3,4]. Among these methods, EK is known as a promising technique for in situ remediation because it can remove multiple-heavy metals simultaneously [5,6]. Remediation of pollutants in soil by EK is considered to promote the ionization of pollutants and drive ionic pollutants to leave the soil in a designated direction, but this process may be hindered to cause compromised remediation effect. Coupling EK with permeable reactive barriers (PRB), especially when the PRB is installed in an appropriate location, could effectively enhance the remediation result [7,8,9]. As the arsenic could be transferred into and captured by the reactive materials in PRB through electromigration, electrodialysis, and electrophoresis [10,11]. Researches have shown that the arsenic removal of the combined EK-PRB was 1.6–2.2 times greater than that of EK alone [12]. Moreover, heavy-metal removal was also dependent on the type of PRB materials [13]. Compared with ion exchange membrane, Fe(0), and activated carbon, the layered double hydroxide (LDH) takes advantage of the high capacity of anionic exchanging.

LDH is classified as a clay consisting of stacked positive layers separated by an interlamellar region constituted of anions and water. Its general formula is M_1−x_^2+^M_x_^3+^(OH_−_)_2_]^x+^(A^n−^)_x/n_·mH_2_O, where M^2+^ is a divalent metal, M^3+^ is a trivalent metal and A^n^_−_ is an anion, the M^2+^/M^3+^ ratio is 0.1 ≤ x ≤ 0.5 molecules [14]. The LDH materials exhibit special structure and function such as replaceable intercalated anions and chemical multi-functionality [15], which enables them to be used as catalysts [16] and adsorption materials [17]. For example, Soltani et al. synthesized the hierarchical LDH/MOF nanocomposite that can be applied as a promising adsorbent for the simultaneous removal of toxic dyes and heavy metals from wastewater [18]. Hu et al. reported that the Ni/Fe-LDH nanosheet/carbon fiber adsorbent had a much higher adsorption capacity for the elimination of anionic dyes than that of pure carbon fibers [19]. Xu et al. successfully remediated Cr-contaminated soils by EK coupled with Ca/Al-LDH PRB [20], suggesting coupling EK with LDH PRB has a great potential for contaminated soil remediation. Since arsenic is strongly bound to Fe, Al, and Mn oxides in soils, resulting in poor mobility and low removal rate [21,22], it is necessary to identify the main influencing factors to improve the arsenic removal rate in EK-PRB operation.

In this study, EK-PRB with Fe/Mn/C-LDH materials as PRB filler was employed to remediate the arsenic-contaminated soils. The effects of the operation conditions such as electric field strength, PRB position and moisture content were systematically investigated. To reveal the removal mechanism of arsenic from contaminated soil by EK-PRB, the Fe/Mn/C-LDH after EK and reaction was characterized by various techniques including scanning electron microscope, X-ray diffractometer, Fourier transform infrared, and zeta potential analyzer.

## 2. Materials and Methods

### 2.1. Materials

#### 2.1.1. Soil

The soil used in this study was collected from a pollution-free site near a lead-zinc mining area in Hechi City, Guangxi Zhuang Autonomous Region. The sampling site belongs to the Karst geomorphology area, located in the Huanjiang mountain area, belongs to the subtropical monsoon climate area, and has abundant rainfall. The soil samples were taken at a sampling depth of 5–20 cm, and dried naturally after removal of stones and plant residues, named latosol, which is classified as kandiudults of udults by the US soil taxonomy. The dried soil was sieved through a 20-meshes sieve and thoroughly mixed with a certain amount of arsenic solution (0.87 g NaAsO_2_ per 500 mL deionized water). The target arsenic-contaminated soil was obtained after 1-month aging and culture with a moisture content of about 35%. The physical and chemical properties of soil before and after pollution are shown in Appendix A.

#### 2.1.2. Preparation of Fe/Mn/C-LDH

The Fe/Mn/C-LDH material was prepared by using bamboo as bio-templates. The bamboo slices were boiled in 5% dilute ammonia water at 100 °C for 6 h, dried at 80 °C for 24 h and then annealed in a muffle furnace at 600 °C for 3 h. Then, the bamboo charcoals were soaked in concentrated HNO_3_ at 110 °C for 2 h, washed with deionized water and dried at 80 °C to obtain bamboo charcoal bio-templates. Afterwards, the bamboo charcoal bio-templates were immersed in deionized water, a mixed solution of Fe and Mn (Fe/Mn molar ratio = 2:1) with a total concentration of 1 M was added. Subsequently, the solution pH was adjusted to 11.5 by the mixture solution of 3.2 M NaOH and 0.1 M Na_2_CO_3_. Finally, the mixed solution was continuously stirred for 2 h, aged at 60 °C for 24 h, filtered and freeze-dried to obtain Fe/Mn/C-LDH materials.

### 2.2. Construction and Operation of EK-PRB Device

The experiment was carried out in an EK-PRB remediation device, which was mainly composed of a polymethyl methacrylate chamber (300 mm × 100 mm × 50 mm), including electrode, direct current (DC) power supply and PRB filler (Appendix A). Graphite plates (100 mm × 10 mm × 50 mm) were served as both anode and cathode, and the Fe/Mn/C-LDH material was used as the PRB filler. For each 96 h run, 600 g of the arsenic-contaminated soil was placed in the soil chambers. The voltage, PRB location and moisture content were varied to investigate the effects of the operating condition.

### 2.3. Analysis Methods

#### 2.3.1. Soil Physical and Chemical Properties

After each run, the treated soil was subsampled, dried, ground and sieved through a 100-mesh sieve. The pH and electrical conductivity (EC) of the treated soil were measured by pH and EC meters (pHS-3E, Shanghai, China), respectively, from extract prepared by shaking samples with deionized water with a soil/water ratio of 1:2.5 (pH) and 1:5 (EC). The redox environment of arsenic in soil was described by measuring the redox potential of soil pore water [23].

#### 2.3.2. Analysis Method for Arsenic Content

The leachability of arsenic in the treated soils was estimated by employing the in vitro toxicity characteristic leaching procedure (TCLP) [24]. Specifically, 2.5 g of dry soil was mixed with 50 mL TCLP extract, and shaken at 25 °C for 18 ± 2 h. Besides, to determine the total and residual arsenic content, 0.2 g of dry soil was digested in 10 mL of aqua regia in a boiling water bath for 2 h. The arsenic concentration was analyzed using an inductively coupled plasma spectrometer (Optima 7000DV, Waltham, MA, USA) [25].

#### 2.3.3. Analysis of Arsenic Speciation

A modified four-step extraction procedure (BCR, European Community Bureau of Reference) was used to examine the chemical species of arsenic in soil [26]. The specific steps are shown in Appendix A.

#### 2.3.4. Characterization Methods

Materials characterizations are shown in the Appendix A.

## 3. Results and Discussion

### 3.1. Structural Performances of Materials

The SEM images of bamboo charcoal (BC) after being treated with dilute ammonia solution are shown in Figure 1A,B. The BCs exhibited a long multi-layer structured tubular shape with a large number of interconnected pores. The vascular bundles of BC were composed of a large number of fiber holes, ladder ducts, threaded holes and ring tubes. Besides, a large number of fiber tubes in BC were divided into different unit structures with various lengths, with nanopores distributed in each unit structure. Through the pretreatment and extraction operation of soaking in 5% dilute ammonia boiling water bath for 6 h, on one hand, some impurities in the gap of BC were removed, and the connection between various pipelines was destroyed, leading the internal pipelines of BC in an open connected state. On the other hand, the extraction process made the plant fiber and hemicellulose of BC expand, and the protein macromolecules, amino acid small molecules, lignin and a small part of resin oil dissolved, which provide a skeleton for material loading in the later stage. This multi-layer complex porous structure made BC a good template for material loading. Due to its porous structure and adsorption capability, BC could adsorb metal substances to make a good composite material after modification.

Figure 1C,D shows the SEM images of Fe/Mn/C-LDH. It maintained the original porous structure of BC. A large number of LDHs with hexagonal and orthorhombic crystal structures were uniformly distributed on the surface and inside the pores of BC, which were typical SEM morphology characteristics of layered metal oxides. The subsequent characterizations confirmed that the loaded material was Fe/Mn-LDH. In conclusion, Fe/Mn/C-LDH composite PRB filler was successfully prepared.

Figure 2 and Appendix A show that a large number of C, O, Fe and Mn elements were detected on the surface of the material. The mass percentage of Fe and Mn element was 37.85%, the weight ratio of Fe element and Mn element was 0.54 from the EDS analysis result in Appendix A, which is close to Fe/Mn metal ion ratio 1:2 used in the experiment. It could be inferred that Fe/Mn/C-LDH was successfully loaded in BC. Moreover, it could be seen that Fe/Mn/C-LDH composite had abundant oxygen-containing functional groups from the high mass percentage of C and O, which could not only improve the modification performance of the material, but also greatly improve the adsorption capacity of the material. The results indicated that as-prepared Fe/Mn/C-LDH could be used as PRB filler for arsenic removal from soil by EK-PRB.

Figure 3 shows the XRD patterns of BC and Fe/Mn/C-LDH. Besides the diffraction peaks of BC with wide shoulder width appearing at 2θ = 23.859° and 43.132°, there were no other diffraction peaks. The widening of the characteristic diffraction peak as well as the absence of other diffraction peaks was ascribed to the low crystallinity of the BC material. In comparison, pronounced diffraction peaks were observed at 2θ = 24.180°, 31.330°, 37.440°, 41.340°, 45.140°, and 51.609 for Fe/Mn/C-LDH. The presence of emerging diffraction peaks was attributed to the LDH by comparing with standard PDF card. The abundant functional groups on the surfaces of BC may react with Fe and Mn elements to form carbonate. Thus, diffraction peaks of MnCO_3_ were observed in the XRD pattern, such as at 2θ = 31.330°.

As shown in Figure 4, the O-H stretching vibration peaks of adsorbed water in 3367~3400 cm^−1^, and the H-O-H bending vibration peaks that appeared in 1558.30 cm^−1^ and 1712.35 cm^−1^ were observed for BC, Fe/Mn/C-LDH, and Fe/C. The absorption peak of BC at 1238.35 cm^−1^ was the stretching vibration peak of C=O double bond. The absorption peaks of Fe/Mn/C-LDH relative to BC at 863.17 cm^−1^ and 592.13 cm^−1^ were the characteristic vibrational absorption peaks of Mn-O and Fe-O of LDH metal laminates. It showed that Fe and Mn were successfully loaded on BC, verified that Fe/Mn/C-LDH composite PRB filler had been successfully prepared. Similarly, the absorption peak of Fe/C relative to BC at wave number 554.61 cm^−1^ was assigned to Fe-O stretching vibration, indicating that Fe was successfully loaded on BC to obtain Fe/C composite.

As shown in Figure 5, the zero-point potential of Fe/Mn/C-LDH was at pH = 4.38. The results showed that when the pH value of the solution system was less than 4.38, the surface of Fe/Mn/C-LDH composite was positively charged, which was favorable for the adsorption of anionic pollutants. Similarly, when the pH value of the solution was greater than 4.38, the surface of Fe/Mn/C-LDH composite was negative charged, favorable for the adsorption of cationic pollutants. Due to the arsenite ions in the soil being negatively charged, lower pH value (<4.38) was favorable for the adsorption of arsenic by Fe/Mn/C-LDH.

### 3.2. The Main Influence Factors of EK-PRB

In the process of remediation of arsenic-contaminated soil by EK-PRB, due to the influence of polarization, the location of PRB will directly affect the pH value of its reaction environment, and voltage and moisture content will directly affect the generation of polarization, polarization degree and electric driving force. Therefore, the voltage gradient, PRB position, moisture content and PRB filler type were the key factors affecting the remediation of arsenic-contaminated soil by EK-PRB.

#### 3.2.1. Voltage Gradient

The principle of an electrically driven enhanced permeable reaction wall for the remediation of arsenic-contaminated soil is to generate DC electric field by applying a voltage to the soil using an external power supply. In an electrically driven system, the voltage directly affects the driving capability. However, excessive voltage would promote the generation of by-products and increase energy consumption. To investigate the effects of initial voltage gradients on arsenic remediation, voltage gradients of 1, 2, and 3 V/cm were used in the EK-PRB experiments, respectively.

The leaching toxicity concentration and removal rate after EK-PRB remediation were illustrated in Figure 6A and Appendix A. At the voltage gradient of 1, 2 and 3 V/cm, the leaching toxicity concentrations decreased from 96.92 mg/kg to 6.38, 3.96 and 1.55 mg/kg, respectively, and the maximum leaching toxicity removal rates were 93.42%, 95.71% and 98.40%, respectively. It is worth noting that the average leaching toxicity removal rates were 58.83%, 86.11% and 84.49%, respectively. The results indicated that the leaching toxicity removal rate near the anode was the highest at the voltage gradient of 3 V/cm, but the average leaching toxicity removal effect was the best under the voltage of 2 V/cm when both the power consumption and remediation effect were taken into account. In addition, due to the positive correlation between the electric driving capacity and the voltage gradient, the leaching toxicity removal rate is the lowest under the condition of 1 V/cm.

The residual concentrations of total arsenic in soil at different voltage gradients after 96 h treatment were compared in Figure 6B. The concentrations of total arsenic were 418.72, 396.02, and 367.19 mg/kg, respectively. Therefore, the poor electric driving capability under the low voltage gradient led to inadequate arsenic migration ability in the soil, resulting in more arsenic residue in the soil.

Properly increasing the voltage could improve the migration effect of arsenic to a certain extent. However, from the results of Figure 6B, under the condition of 2 V/cm and 3 V/cm, the difference of arsenic removal as well as total arsenic residue in soil after EK-PRB remediation is not significant, consistent with the analysis conclusion of Figure 6A. Increasing applied voltage resulted in excessive consumption of interstitial water in soil due to enhanced water electrolysis, which affected the ionization rate of arsenic in soil. However, the mechanism of the process needs to be further studied.

The distribution of soil pH at different voltage gradients after EK-PRB treatment was shown in Figure 6C. For the distance from the cathode being 5 cm, the soil pH after EK-PRB were 7.98, 7.86, and 5.65, respectively. It showed that the higher the applied voltage, the lower the pH value in soil. As the high voltage would improve the ability of electric drive and the degree of electrolytic water, which would enhance the release of H^+^ and decrease pH value in soil. Notably, compared with 1 V/cm and 2 V/cm, water electrolysis under 3 V/cm become intense, resulting in rapid decline of pH in the soil in the middle of the soil trough apart from the cathode 3–7 cm.

The variation of EC in soil is shown in Figure 6D. The difference between the voltage gradient of 2 V/cm and 3 V/cm was not obvious. The EC in soil near the anode was 2365.5 us/cm when the voltage gradient was 1 V/cm, which was 654 us/cm lower than that of 2 V/cm. It could be seen that EC increased with the increase of voltage gradient in the low voltage range, which is consistent with the change of current. However, when the voltage gradient was greater than 2 V/cm, the improvement effect of voltage gradient on EC was not obvious. It means that the voltage gradient condition of 2 V/cm, has met the requirement to dissolve the arsenic-containing solids while not enough to driving the ionization of arsenic in soil. Therefore, it is economical to use voltage gradient of 2 V/cm to drive the dissolution of arsenic-containing solids.

As shown in Figure 6E, for all applied voltages, the current increased until a peak of current appeared, which was 0.06 A, 0.09 A and 0.12 A, respectively, and then decreased rapidly and finally stabilized. Generally speaking, this phenomenon could be explained by the releasing rate of total dissolved ion in soil, and the total dissolved ions in soil were critical to maintaining the current. In detail, a large number of H^+^ produced by water electrolysis moved to the cathode, leading to soil acidification and promoting the release of ions in soil [27]. Hence, the current density would rise at the inception phase. However, with the increase of treatment time, the concentration of mobile ions in soil decreased, which resulted in a decrease in current density [28]. Besides, the current increased as the applied voltage gradient increased, consistent with Ohm’s law. In addition, it can be seen that the current attenuation was more severe under the condition of 1 V/cm. Combined with Figure 6C analysis, the current attenuation was greatly affected by pH in the soil, which may be caused by the hydroxide precipitates formed in the soil under higher pH.

#### 3.2.2. PRB Position

In the operation of EK-PRB, the soil near the cathode was alkaline and the anode was acidic due to the electrolysis of electrolyte. The charge on the PRB surface was different when the PRB was placed in different positions, which influenced the removal effect of PRB on arsenic. Thus, in order to investigate the effects of PRB positions on arsenic remediation, three different positions were applied in the EK-PRB experiment with PRB placed in the middle, near the anode and near the cathode, respectively.

The leaching toxicity removal rate after EK-PRB remediation was illustrated in Figure 7A. When PRB was placed near cathode or anode, the leaching toxicity concentrations decreased from 106.68 to 11.57 and 7.31 mg/kg, respectively. When PRB was placed in the middle, the leaching toxicity concentration decreased from 92.38 to 3.96 mg/kg. Additionally, the maximum soil leaching toxicity removal rates were 89.15%, 95.71% and 93.15%, respectively. The average leaching toxicity removal rates were 62.29%, 88.03% and 85.31%, respectively. The best removal rate was obtained with PRB placed in the middle, and the removal rate with PRB placed near the anode was better than that placed at the cathode. Combined with Figure 7C analysis, this phenomenon occurs because Fe/Mn-LDH, Fe/Mn-LDH has the best adsorption performance for arsenic ions at pH = 5–8 (the corresponding PRB position is in the middle of the soil trough). At the same time, under this condition, the leaching toxicity removal rate is the highest.

The distribution of arsenic residue when PRB was placed at different locations after EK-PRB treatment was shown in Figure 7B. When PRB was placed at anode or cathode, the concentrations of residual arsenic were 396.02 and 372.71 mg/kg. However, the concentration of residual arsenic was 344.04 mg/kg when PRB was placed in the middle. By comparison, the removal effect of arsenic with PRB placed in the middle was better than that placed at anode or cathode. The result might be attributed to the fact that anode was acidic during EK-PRB operation, resulting in partial dissolution of PRB filler placed near the anode. This inference was consistent with the result of soil pH change. Therefore, in general, the removal effect was best when PRB was placed in the middle, consistent with the result presented in Figure 7A.

The distribution of soil pH after EK-PRB remediation was shown in Figure 7C. A general trend of low pH near the anode and high pH near the cathode was found for this EK-PRB experiment. The pH distribution in the soil was determined by the mobility of H^+^ and OH− as well as the initial concentration of other ions in the system. The consumption of OH− in precipitation, the surface complexation and adsorption by the soil, and the formation of complexes in the bulk solution were other factors affecting the pH distribution in the EK-PRB system [29]. The acid generated at the anode reservoir flushed across the soil specimen, which led to lower the pH of soil to around 2.46–3.05 near the anode in all cases. At the cathode side, the migration of OH− toward the anode would cause the soil pH up to 9.04–9.52. Results showed that no evident relationship between soil pH and PRB positions, which was consistent with previous research [12].

The variation of EC in soil is shown in Figure 7D. The changing trend of EC was basically the same in spite of where PRB was placed for remediation. The EC of the PRB placed in the middle was higher near the anode, possibly due to the ions in the PRB filler participating in the process of electric drive, which made the EC slightly higher than the other two groups. The EC of the three experimental comparison groups was that the EC at the anode was much greater than the cathode, which was due to the fact that the rate of hydrogen ion produced by anode electrolysis was 1.8 times that of hydroxyl ion produced by cathode electrolysis [30].

The variation of current with time at different PRB positions is shown in Figure 7E. The current recorded with different PRB positions showed a similar trend, increased to a park firstly, then decreased rapidly, and finally stabilized. When PRB was placed near the cathode, the current density increased rapidly at the beginning. The reason might be that the cathode electrolyte was acidic, which could promote the dissolution of ions in PRB materials, and increase the current density. However, the current density decreased gradually after 24 h. It might be that OH^-^ produced by anode reacted with ions in soil to form compounds. Additionally, the clogging of soil pores by complexing compounds resulted in a low current density at final.

#### 3.2.3. Moisture Content

Soil moisture content would affect the correlation coefficients such as current and electrodialysis, which indirectly affected the ability of electrically driven removal of arsenic in soil. The increase of soil moisture content could not only improve the porosity of soil, but also drive the electrolyte in soil through water molecules to promote the electric driving effect. To investigate the effects of soil moisture content, the soil moisture content was adjusted to 25%, 35% and 45% in the EK-PRB experiments, respectively.

As shown in Figure 8A, the leaching toxicity concentration decreased from 84.14, 92.38, and 95.14 mg/kg to 4.93, 3.96, and 2.88 mg/kg, respectively. Additionally, the maximum leaching toxicity removal rate was 94.78%, 95.71%, and 96.97%, respectively. The average leaching toxicity removal rates were 72.54%, 88.03% and 69.64%, respectively. It could be seen obviously that when the moisture content was 35%, the leaching toxicity removal effect was the best as a whole. Too high and too low moisture content are not conducive to the improvement of leaching toxicity removal rates. If there is too much water in the soil, the flow of water may affect the directional driving effect of EK, while too low water content may be compromise the effect of EK on the migration of arsenic ions, resulting in poor leaching toxicity removal effect.

The distribution of residual arsenic after EK-PRB remediation is illustrated Figure 8B. The concentrations of residual arsenic decreased to 381.75, 396.02 and 375.84 mg/kg, respectively. It showed that soil moisture content had slight effect on the removal of arsenic. In the process of electrokinetic remediation, the migration of pollutants was positively correlated with soil moisture content, and the electroosmotic flow would increase with the increase of soil moisture content. The mobility of arsenic was greater in soil with 35% moisture content [31]. Thus, it was considered that the optimum moisture content in this study was 35%.

After 96-h EK-PRB treatment, the soil pH profiles at different soil sampling points are illustrated in Figure 8C. For all moisture content, the final pH value of the cathode was alkaline and the anode was acidic. The pH at cathode increased from 7.43 to about 9.6, and the pH decreased from 7.43 to about 2.2. Thus, the moisture content had little relation with the change of soil pH.

The change of EC at each sampling point of soil after remediation under different moisture content is shown in Figure 8D. There was a positive correlation between the initial soil moisture content and EC. The EC of soil with 35% and 45% moisture content was significantly higher than that with 25% moisture content. Besides, the EC of the soil near the anode was much greater than that near the anode. The reason was that H^+^ produced by the anode had high EC, which made the EC near the anode side high, resulting in an increasing trend of EC from cathode to anode.

The relationship between current density and moisture content is shown in Figure 8E. The current density was only 0.01 A when the soil with 25% moisture content at the inception phase. Weng et al. found that there was a certain correlation between soil moisture content and current [32]. Besides, the peak of current was 0.074 A when the soil had 25% moisture content. By comparison, the peak of current density was higher when the soil had 35% and 45% moisture content. That was due to the direct influence of moisture on the conductivity of soil. Later, the current gradually decreased and finally stabilized, which might be correlated with the depletion of mobile ions in the soil or electrode polarization, etc. [33].

### 3.3. Mechanism Analysis

Based on the above various characterization analyses, the possible arsenic removal mechanisms in soil by EK-PRB are shown in Figure 9. In the EK-PRB system, arsenic existed in soil solution in the form of anion. Additionally, exchangeable and carbonate fractions of As [10] could exchange ions with −OH or −OH_2_ on the surface of soil particles like Fe, Al and Mn oxides, so as to be strongly adsorbed on the coordination site of metal ions [34,35]. The pH value of soil changed under the action of electric field. Higher pH was observed near the cathode, which caused the metal precipitation near the cathode, thereby blocking soil pores and hindering the remediation process [36]. Additionally, it was more conducive to the desorption of arsenic on soil particles with the increase of pH value. Besides, electric current had a dominant effect on temperature changes, which caused the soil moisture content to rise with the decreasing of pH [37]. The increase of pH could also suppress the formation of arsenic compounds such as H_2_AsO_4_^−^ or H_3_AsO_4_, which declines the efficiency of electrokinetic remediation. Various arsenic ions in soil interstitial water separated from the soil and entered the aqueous phase. Meanwhile, electroosmotic flow and electromigration were also carried out under the influence of electric drive. Arsenic ions migrated to the anode through electromigration, and to the cathode with the action of electroosmotic flow, the arsenic distribution was concordant with the voltage drops distribution, which indicated the voltage loss played a critical role in arsenic migration [38]. In this process, arsenic ions would enter PRB and then react with Fe/Mn/C-LDH through a series of reactions, such as electrostatic adsorption, ion exchange, surface functional group complexation, physical adsorption, etc., to achieve better removal of arsenic from soil.

## 4. Conclusions

A novel remediation technique by coupling electrokinetics (EK) with the PRB of Fe/Mn/C-LDH composite was applied for the remediation of arsenic-contaminated soils. The Fe/Mn/C-LDH PRB fillers synthesized by using bamboo as a template retained the porous characteristics of the original bamboo, the mass percentage of Fe and Mn element was 37.85%, and the mass ratio of Fe element and Mn element was estimated to be 0.54 from the EDS result. The placement of PRB of Fe/Mn/C-LDH in the middle was a feasible choice in this work, with the maximum and average soil leaching toxicity removal rates being 95.71% and 88.03%, respectively. That was due to the fact that the pH value in the middle of the soil trough was 5–8, which was favorable for the adsorption of arsenic onto the PRB filler Fe/Mn-LDH. When the voltage gradient was 2 V/cm, both the arsenic removal and economic aspects were optimal, with the maximum and average soil leaching toxicity removal rates being 98.40% and 84.49% of the result under the voltage gradient was 3 V/cm, respectively. Besides, the soil moisture content had a negligible effect on the removal of arsenic, but a slight effect on leaching toxicity. The best leaching toxicity removal rate was achieved when the soil moisture content was 35%, neither higher nor lower moisture in the range of 25–45%, which was conducive to the improvement of leaching toxicity removal rates.

## Figures and Tables

**Figure 1 ijerph-19-04389-f001:**
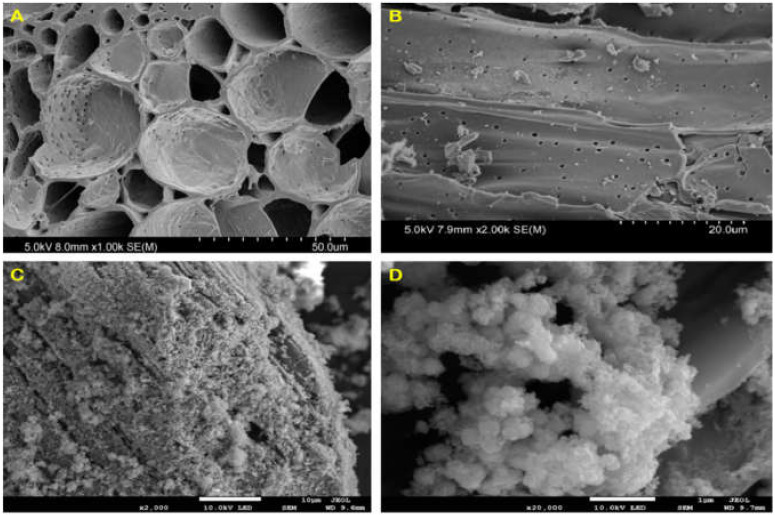
The SEM images of BC (**A**,**B**) and Fe/Mn/C-LDH (**C**,**D**).

**Figure 2 ijerph-19-04389-f002:**
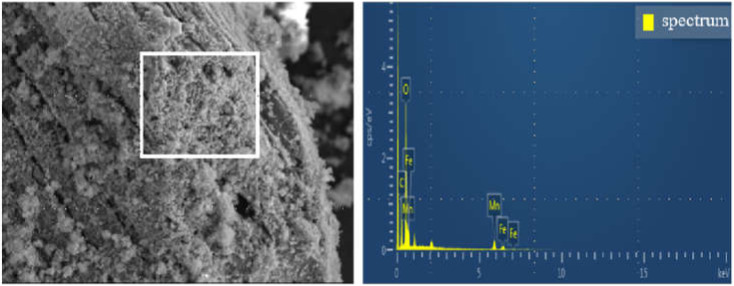
The EDS spectrum of Fe/Mn/C-LDH.

**Figure 3 ijerph-19-04389-f003:**
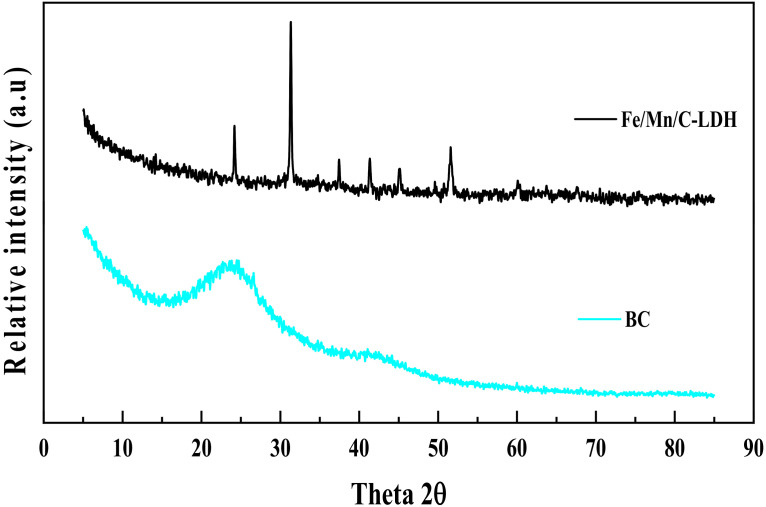
The XRD patterns of BC and Fe/Mn/C-LDH.

**Figure 4 ijerph-19-04389-f004:**
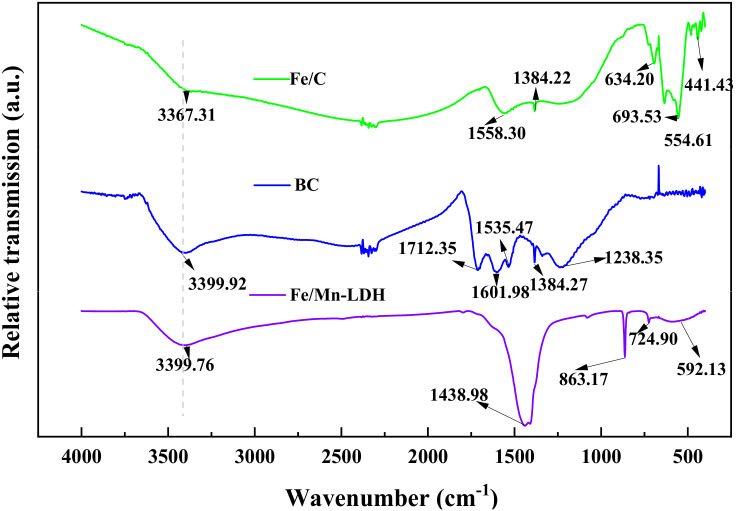
The FT-IR spectra of Fe/C, BC and Fe/Mn/C-LDH.

**Figure 5 ijerph-19-04389-f005:**
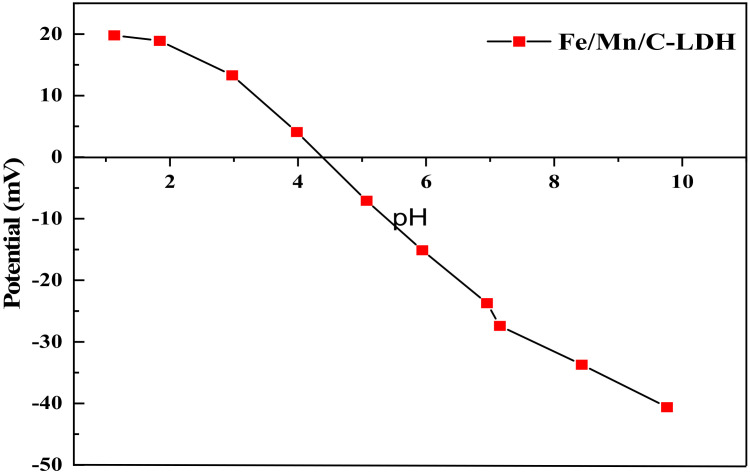
The Zeta potential of Fe/Mn/C-LDH.

**Figure 6 ijerph-19-04389-f006:**
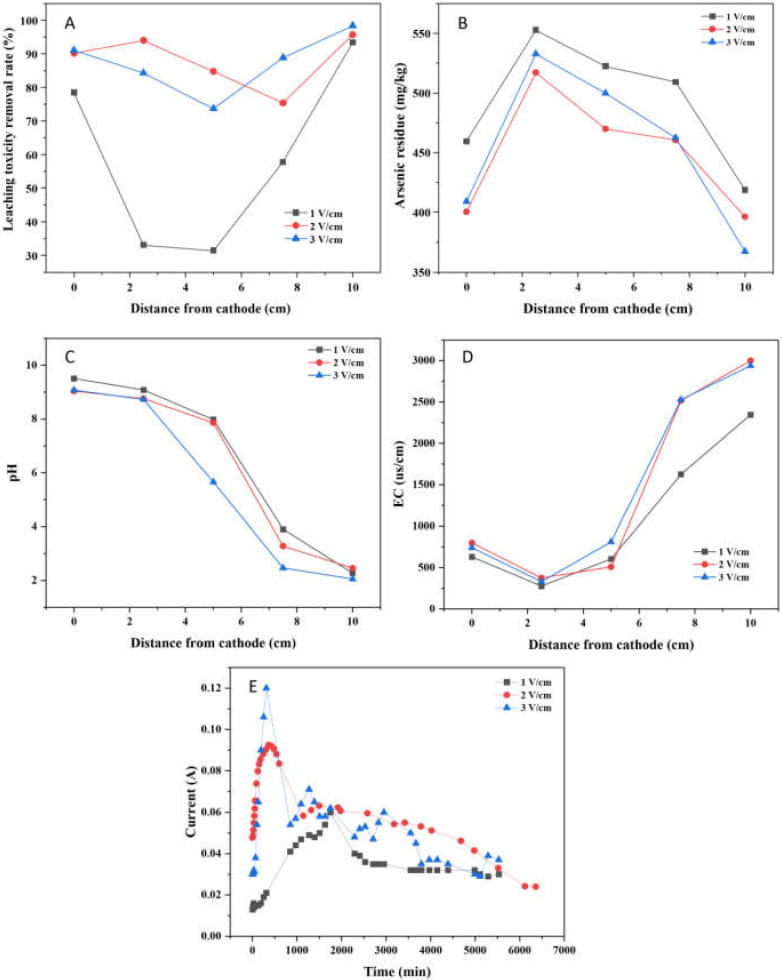
Effect of voltage gradient for the remediation of arsenic-contaminated soil by EK-PRB (Leaching toxicity removal rate (**A**), arsenic residue (**B**), Ph (**C**), EC (**D**) and current (**E**)).

**Figure 7 ijerph-19-04389-f007:**
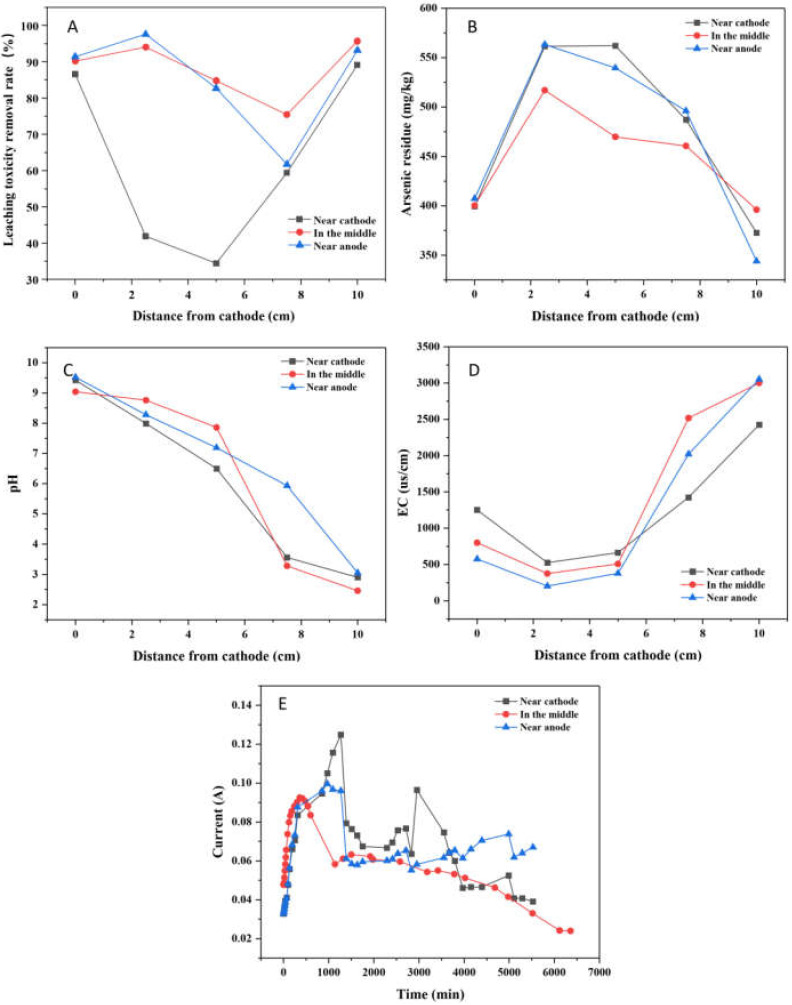
Effect of PRB position for the remediation of arsenic-contaminated soil by EK-PRB (Leaching toxicity removal rate (**A**), arsenic residue (**B**), pH (**C**), EC (**D**) and current (**E**)).

**Figure 8 ijerph-19-04389-f008:**
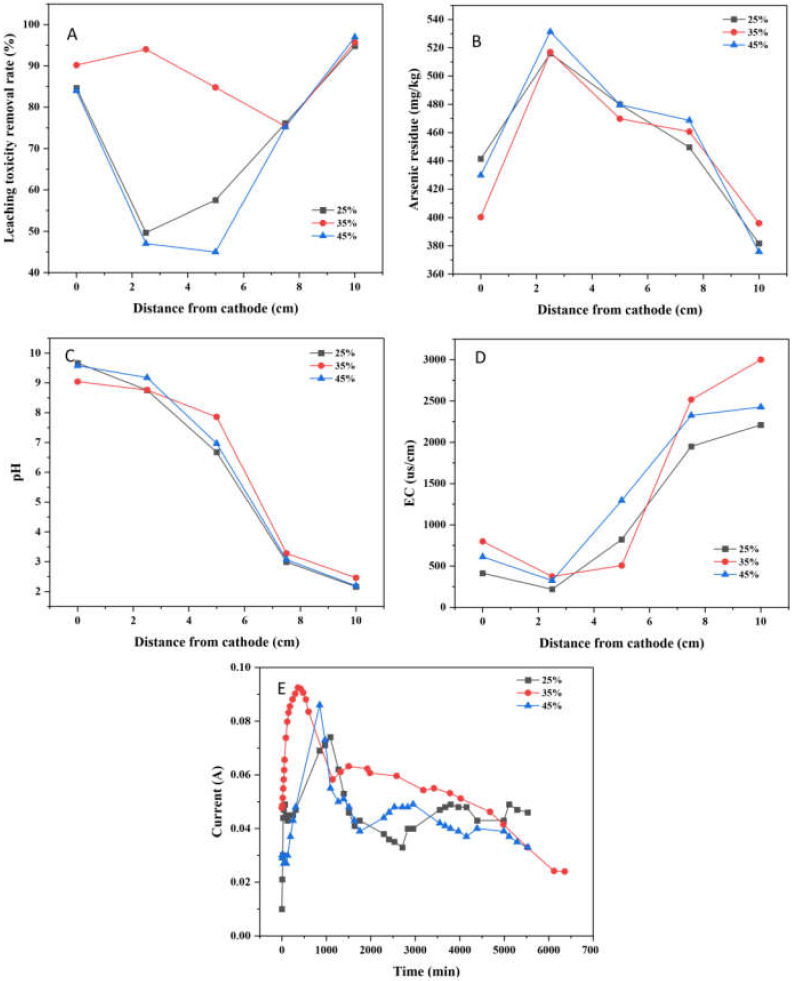
Effect of moisture content for the remediation of arsenic-contaminated soil by EK-PRB (Leaching toxicity removal rate (**A**), arsenic residue (**B**), pH (**C**), EC (**D**) and current (**E**)).

**Figure 9 ijerph-19-04389-f009:**
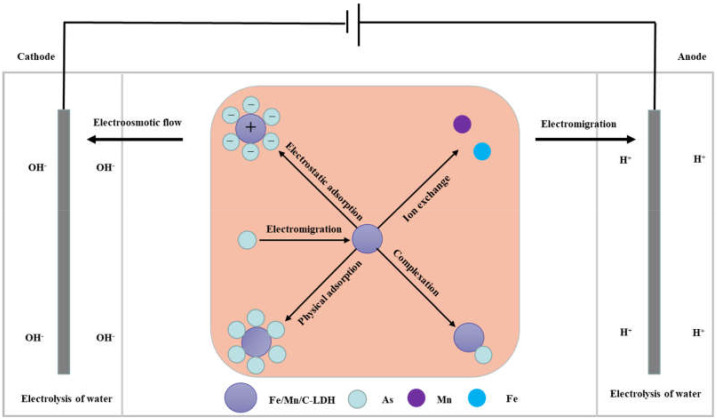
Schematic diagrams of arsenic removal mechanisms by EK-PRB.

## Data Availability

Not applicable.

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
