# Peer review of "Effective Remediation of Arsenic-Contaminated Soils by EK-PRB of Fe/Mn/C-LDH: Performance, Characteristics, and Mechanism"

_ijerph, 2022, doi:10.3390/ijerph19074389_

Round 1

Reviewer 1 Report

In this manuscript, a device of EK coupled with the PRB of Fe/Mn/C-LDH was designed, which was experimentally used for remediating As-contaminated soils. The authors mainly studied the influences of the voltage gradient, PRB position, moisture content of soil and PRB filler types on the removal rate of As in soils. They found the removal rate of As in the contaminated soils reaches more than 95% under the optimum conditions of the device. The results seem encouraging.

Scientifically, the mechanism of As removal by the EK-PRB is not a new idea. There are not clear findings in the conclusions. English expression of the whole manuscript is not so good and should be re-written completely.

Line 16: It should be “remediation of arsenic-contaminated soils”.

Line 17-18: It should be “The influences of ---- on the removal rate of arsenic in the contaminated soils were studied.”

Line 19: The removal rate of “toxicity” or “As”?? The expression should be revised completely.

Line 388: Mechanism of arsenic removal??

Line 389-392: it is commonly known.

Line 411-414: Delete these useless words.

Line 411-421: The conclusions do not include your main research findings, and should be re-rewritten.

Author Response

Reviewers' Specific comment

Our response and revision

(Revise according to reviewers’ advices)

Scientifically, the mechanism of As removal by the EK-PRB is not a new idea. There are not clear findings in the conclusions. English expression of the whole manuscript is not so good and should be re-written completely.

Rewrite the manuscript:

We Rewrite the full text to enhance the English expression of the whole manuscript.

Line 16: It should be “remediation of arsenic-contaminated soils”.

Revise the following sentence in the text:

…composite was applied for remediation of arsenic - contaminated soils.

Line 17-18: It should be “The influences of ---- on the removal rate of arsenic in the contaminated soils were studied.”

Revise the following sentence in the text:

The influences of electric field strength, PRB position, moisture content and PRB filler type on the removal rate of arsenic from the contaminated soils were studied.

Line 19: The removal rate of “toxicity” or “As”?? The expression should be revised completely.

Revise the following sentence in the text:

“The setting of PRB of Fe/Mn/C-LDH placed in the middle is a feasible option, with the maximum and average soil leaching toxicity removal rates of 95.71% and 88.03%, respectively. When the electric field strength was 2 V/cm, both the arsenic removal rate and economic aspects were op-timal.”

Line 388: Mechanism of arsenic removal??

Revise the following sentence in the text:

the possible arsenic removed mechanism in soil by EK-PRB was shown in Fig.9.

Line 389-392: it is commonly known.

Delete the following sentence in the text:

“the migration process of arsenic was usually simply described as follows: arsenic ions firstly desorbed from soil particle to soil-water, and then transported towards the anode under an electrostatic field, and were finally captured by PRB. In detail”

Line 411-414: Delete these useless words.

Delete the following sentence in the text:

“The effects of voltage gradient, PRB position, moisture content, and PRB types on the remediation of arsenic contaminated soil by EK-PRB and its mechanism were studied.”

Line 411-421: The conclusions do not include your main research findings, and should be re-rewritten.

Rewrite the conclusions in the text:

A novel remediation technique by coupling electrokinetics (EK) coupled with the PRB of Fe/Mn/C-LDH composite was applied for the remediation of arse-nic-contaminated soils. The Fe/Mn/C-LDH PRB fillers synthesized by using bamboo as a template retained the porous characteristics of the original bamboo, the mass per-centage of Fe and Mn element was 37.85%, and the mass ratio of Fe element and Mn element was estimated to be 0.54 from the EDS result. The placement of PRB of Fe/Mn/C-LDH in the middle was a feasible choice in this work, with the maximum and average soil leaching toxicity removal rates being 95.71% and 88.03%, respectively, That was due to the fact that the pH value in the middle of the soil trough was 5-8, which was favorable for the adsorption of arsenic onto the PRB filler Fe/Mn-LDH. When the voltage gradient was 2 V/cm, both the arsenic removal and economic aspects were op-timal, with the maximum and average soil leaching toxicity removal rates being 98.40% and 84.49% of the result under the voltage gradient was 3 V/cm, respectively. Besides, the soil moisture content had negligible effect on the removal of arsenic, but slight ef-fect on leaching toxicity. The best leaching toxicity removal rate was achieved when the soil moisture content was 35%, neither higher nor lower moisture in the range of 25-45%, was conducive to the improvement of leaching toxicity removal rates.

Reviewer 2 Report

This paper is interesting, comprehensive and adequate. This paper deals with the development of a device of electrokinetics coupled with the permeable reactive barriers of Fe/Mn/C-LDH composite for remediation of arsenic contaminated soils.

Strengths - provide a comprehensive study to evaluate the remediation effectiveness of arsenic contaminated soils by using a combination of electrokinetics and home made permeable reactive barriers. Parameters such as voltage gradient, PRB position, moisture content and PRB filler types were studied. Several methods such as scanning electron microscope, x-ray diffractometer, fourier transform infrared and zeta potential analyzer were used to analyse Fe/Mn/C-LDH material.

Weakness ­- The applicability to real situations. What happens in the presence of other heavy metals, or plants or smaller moisture percentages?

Abstract – the information given is adequate and concise.

1.Introduction – The aim of paper is adequately exploited.

2.. Materials and Methods – adequate

Lines 74-75 How was the arsenic contaminated soil prepared? In which chemical forms was it added to the soil?

2.2 Fig 1S should have a scale with the dimensions of each chamber and electrodes.

Table S1 should have the EC values and the redox potential values.

Equipment, experimental design and sampling collection – adequate

Are there reference materials available? If there are why not use them?

3. Results and discussion- Adequate discussion well documented by tables and figures.

Lines 148 and table S3. How was mass percentage of Fe/Mn/C/O determined?

Line 210-211 You state “the leaching toxicity concentrations decreased from 96.92 to 210 6.38 and 1.55 mg/kg, respectively.” How do you get to these values, in Table S1 you have as target arsenic concentration a value of 500 mg/kg and in fig 6A you have %.... Please make it clearer.

Line 218 what is it TAs? Total arsenic?

Line 220 You state that “It showed that the removal rate of arsenic increased with the increase of voltage”. I don’t agree with this statement, you only tested 1, 2 and 3 V/cm, there is a better result if you use 2 instead if 1V/cm, but not if you choose 3 V/cm instead of 2. This sentence should be corrected.

Line 250-251 You state that “It could be seen that EC increased with the increase of voltage gradient, which was consistent with the variation law of current.” Again, I agree with you if we compare 1 and 2 V/cm, but not if you compare 2 and 3 V/cm.

Therefore in my opinion this paper should be accepted with major revision.

Author Response

Reviewers' Specific comment

Our response and revision

(Revise according to reviewers’ advices)

Weakness - The applicability to real situations. What happens in the presence of other heavy metals, or plants or smaller moisture percentages?

Explanation:

Many thanks for the nice comment from anonymous reviewers. The study on the competitive effects of various heavy metals and plants is our next step work.

Lines 74-75 How was the arsenic contaminated soil prepared? In which chemical forms was it added to the soil?

Revise the following sentence in the text:

The dried soil was sieved through 20 meshes and fully mixed with a certain concentration of arsenic solution (0.87g NaAsO2 per 500ml deionized water)

2.2 Fig 1S should have a scale with the dimensions of each chamber and electrodes.

Revise the following sentence in the Fig 1S caption:

(1) Soil chamber, size: 10 cm×5 cm×5 cm (2) Electrolytic chamber, size: 5 cm×5 cm×5 cm (3) Electrode (4) Circulating pump (5) Collecting tank (6) Division plate (7) PRB filler Fe/Mn/C-LDH

Table S1 should have the EC values and the redox potential values.

Are there reference materials available? If there are why not use them?

Explanation:

Many thanks for the nice advice, The EC values and the redox potential values of background soil and arsenic contaminated soil were added in the Table S1.

Lines 148 and table S3. How was mass percentage of Fe/Mn/C/O determined?

Revise the following sentence in the table S3 caption:

1. the tableS3 caption was changed as “Table S3 The weight percentage of elements in Fe/Mn/C-LDH from EDS spectrum”

2. The word “spectrum” in the table S3 was changed to “Weight percentage %”.

3. “The mass percentage of Fe/Mn element was 0.54” was changed as “The mass percentage of Fe and Mn element was 37.85%, the weight ratio of Fe element / Mn element was accounted as 0.54 from the EDS analysis result in Table S3”

Line 210-211 You state “the leaching toxicity concentrations decreased from 96.92 to 210 6.38 and 1.55 mg/kg, respectively.” How do you get to these values, in Table S1 you have as target arsenic concentration a value of 500 mg/kg and in fig 6A you have %.... Please make it clearer.

Revise the following sentence in the text:

The leaching toxicity concentration and removal rate after EK-PRB remediation were illustrated in Fig.6(A) and Table S4. At the voltage gradient of 1, 2 and 3 V/cm, the leaching toxicity concentrations decreased from 96.92 mg/kg to 6.38, 3.96 and 1.55 mg/kg, respectively, the maximum leaching toxicity removal rates were 93.42%, 95.71% and 98.40%, respec-tively. It is worth noting that the average leaching toxicity removal rates were 58.83%, 86.11% and 84.49%, respectively. It means the leaching toxicity removal rate near the anode was the highest at the voltage gradient of 3 V/cm, but average leaching toxicity removal effect was the best under the voltage of 2 V/cm when both the power con-sumption and remediation effect were taken into account. In addition, due to the posi-tive correlation between the electric driving capacity and the voltage gradient, the leaching toxicity removal rate is the lowest under the condition of 1 V/cm.

Add the table S4 in the supplementary:

Table S4 The toxicity characteristic leaching procedure concentration (leaching toxicity) of arsenic on soil after EK-PRB remediation under different voltage gradient conditions

Line 218 what is it TAs? Total arsenic?

Revise the following sentence in the text:

Fig.6(B) compared the residual concentrations of total arsenic in soil at different voltage gradients after 96 h treatment. The concentrations of total arsenic were 418.72, 396.02, and 367.19 mg/kg, respectively.

Line 220 You state that “It showed that the removal rate of arsenic increased with the increase of voltage”. I don’t agree with this statement, you only tested 1, 2 and 3 V/cm, there is a better result if you use 2 instead if 1V/cm, but not if you choose 3 V/cm instead of 2. This sentence should be corrected.

Corrected and revise the following sentence in the text:

The residual concentrations of total arsenic in soil at different voltage gradients after 96 h treatment were compared in Fig.6(B). The concentrations of total arsenic were 418.72, 396.02, and 367.19 mg/kg, respectively. Therefore, the poor electric driving capability under the low voltage gradient led to inadequate arsenic migration ability in the soil, resulting in more arsenic residue in the soil.

Properly increasing the voltage could improve the migration effect of arsenic to a certain extent. However, from the results of Fig.6(B), under the condition of 2V/cm and 3V/cm, the difference of arsenic removal as well as total arsenic residue in soil after EK-PRB remediation is not significant, consistent with the analysis conclusion of Fig.6(A). Increase of applied voltage resulted in excessive consumption of interstitial water in soil due to enhanced, water electrolysis, which affected the ionization rate of arsenic in soil. However, the mechanism of the process needs to be further studied.

Line 250-251 You state that “It could be seen that EC increased with the increase of voltage gradient, which was consistent with the variation law of current.” Again, I agree with you if we compare 1 and 2 V/cm, but not if you compare 2 and 3 V/cm.

Revise the following sentence in the text:

The variation of EC in soil was shown in Fig.6(D). The difference between the voltage gradient of 2 V/cm and 3 V/cm was not obvious. The EC in soil near the anode was 2365.5 us/cm when the voltage gradient was 1 V/cm, which was 654 us/cm lower than that was 2 V/cm. It could be seen that EC increased with the increase of voltage gradient in the low voltage range, which is consistent with the change of current. But when the voltage gradient was greater than 2V/cm, the improvement effect of voltage gradient on EC was not obvious. It means that the voltage gradient condition of 2V/cm, has met the requirement to dissolve the arsenic-containing solids while not enough to driving the ionization of arsenic in soil. So it is economical to use voltage gradient of 2V/cm to drive the dissolution of arse-nic-containing solids.

Reviewer 3 Report

The topic of arsenic contamination in soils is of vital relevance worldwide and devises to improve measurements in these conditions must be supported.

However, the authors are not articulating very well the ideas. The results and discussion section is quite confusing and makes the reading tired. The abstract and the introduction seems to be a series of unrelated thoughts. The abstract is too short with many results presented without much of a context.

The Abstract should provide, in brief, the major motivations to provide the review of this topic. Provide, for instance, some numeric approach, i.e., what is the “size” of this problem? How many people are affected by it? What regions it is covered? And other information…

Basically, the authors should state better to why the scientific community should care about this topic.

Lots of information within the paragraphs, not well constructed.

Line 2: The title is rather too big. The authors should find a better way to express the ideas of the paper in the title.

Line 15: The authors started the phrase directly with “In this work…” It is advised to write the overall challenge to be surpass in the present paper. Conducting an overall opening instead of going direct to the point.

Abstract: the authors must provide clear objectives and/or hypothesis for the study (which is absence now).

Line 34: the sentence “it is worthy to pay our attention” does not sound well (must be improved and re-written).

Line 62: the phrase “In this study, by using…” is too long and not well constructed.

Line 64: The phrase “After reaction…” is also too long and poorly written.

The authors are providing the core information only at the end of phrase, which should be the opposite.

Line 71: “taken from” is too informal. The authors should be using more proper scientific writing.

Line 73: why sampling from 5-20 cm? what is the reason to remove the 0-5 cm soil? And why sampling this depth (any particular reason)? Please specific.

Line 70 to 78: the authors must provide more information about the soil characteristics, such as the complete soil classification (by either the US soil taxonomy or the WRB system). Also provide the geology, geomorphology, and climate characteristics of the field site sampling.

The methods topics 2.1 to 2.3 must go through a detailed English and style correction and better articulating of ideas. The topics seems to be launched as be bullet points in the text.

Line 145: the SEM and EDS are very good.

Line 209: Starting a phrase with “Fig.6(A) illustrated” is not a good way of writing science.

Line 218: similar observation as above for starting the phrase with “Fig.6(B) compared” Please make sure to improve this structure in other parts of the paper (lines 230, 241, 247, and others)

From the topic 3.2 and beyond (for the results and discussion) the reading becomes very tiring. The authors did not make a good job in articulating the ideas.

The figures 6, 7, 8, 9 are very similar and confusing. Better diagrams must be done to better present the data.

Line 411: that is not a nice way to start the conclusions with “In this study…”

The conclusions are too short and seems to be quite similar to what is presented in the abstracted (maybe copied?)

References: 36 references are a very small number for a paper. The authors cited only 1 paper from 2021 and none from 2022. More of the state of the art in the last few years and months must be in the paper.

Author Response

Reviewers' Specific comment

Our response and revision

(Revise according to reviewers’ advices)

However, the authors are not articulating very well the ideas. The results and discussion section is quite confusing and makes the reading tired. The abstract and the introduction seems to be a series of unrelated thoughts. The abstract is too short with many results presented without much of a context.

The Abstract should provide, in brief, the major motivations to provide the review of this topic. Provide, for instance, some numeric approach, i.e., what is the “size” of this problem? How many people are affected by it? What regions it is covered? And other information.

Basically, the authors should state better to why the scientific community should care about this topic.

Lots of information within the paragraphs, not well constructed.

Rewrite the abstract and introduction part to show the major motivations to provide the review of this topic. Add the objectives of the study at the end of the introduction section.

Rewrite the abstract part:

[Context and background] Arsenic is highly toxic and carcinogenic. The aim of the present work is to develop a good remediation technique for arsenic contaminated soil. [Methods] Here, a novel remediation technique by coupling electrokinetic (EK) with the permeable reactive barriers (PRB) of Fe/Mn/C-LDH composite was applied for the remediation of arsenic-contaminated soils. The influences of electric field strength, PRB position, moisture content and PRB filler type on the removal rate of arsenic from the contaminated soils were studied. [Results] The Fe/Mn/C-LDH filler synthesized by using bamboo as a template retained the porous characteristics of the original bamboo, and the mass percentage of Fe and Mn element was 37.85%. The setting of PRB of Fe/Mn/C-LDH placed in the middle is a feasible option, with the maximum and average soil leaching toxicity re-moval rates of 95.71% and 88.03%, respectively. When the electric field strength was 2 V/cm, both the arsenic removal rate and economic aspects were optimal. The maximum and average soil leaching toxicity removal rates were similar to 98.40% and 84.49% of 3 V/cm, respectively. Besides, the soil moisture content had negligible effect on the removal of arsenic but slight effect on leaching toxicity. The best leaching toxicity removal rate was achieved when the soil moisture content was 35%, neither higher nor lower moisture content in the range of 25-45% was conducive to the improvement of leaching toxicity removal rates. The results shown that the EK-PRB technique could effectively remove arsenic from the contaminated soils. Characterizations of Fe/Mn/C-LDH indicated that the electrostatic adsorption, ion exchange, and surface functional group complexation were the primary ways to remove arsenic.

Add the follow section in the introduction part:

In this study, EK-PRB with Fe/Mn/C-LDH materials as PRB filler was employed to remediate the arsenic-contaminated soil. The effects of the operation conditions such as electric field strength, PRB position and moisture content were systematically in-vestigated. To reveal the removal mechanism arsenic from contaminated soil by EK-PRB, the Fe/Mn/C-LDH after EK and reaction was characterized by various tech-niques including scanning electron microscope, x-ray diffractometer, fourier transform infrared, zeta potential analyzer.

Line 2: The title is rather too big. The authors should find a better way to express the ideas of the paper in the title.

Revise the title to express the ideas of the paper:

Effectively remediation of arsenic contaminated soils by EK-PRB of Fe/Mn/C-LDH: Performance, characteristics, and mechanism

Line 15: The authors started the phrase directly with “In this work…” It is advised to write the overall challenge to be surpass in the present paper. Conducting an overall opening instead of going direct to the point.

Rewrite the abstract.

Abstract: the authors must provide clear objectives and/or hypothesis for the study (which is absence now).

Add the clear objectives of the study in the abstract part:

[Context and background] Arsenic in soil is highly toxic and carcinogenic. [Objective] The aim of the present work was to develop a good arsenic pollution remediation technology.

Line 34: the sentence “it is worthy to pay our attention” does not sound well (must be improved and re-written).

Revise the following sentence in the introduction section:

Among these methods, EK is known as a promising technique for in situ remediation because it can remove multiple-heavy metals simultaneously[5,6]. Remediation of pollutants in soil by EK is considered to promote the ionization of pollutants and drive ionic pollutants to leave the soil in a designated direction, but this process may be hindered to cause compromised the remediation effect. Coupling EK with permeable reactive barriers (PRB), especially when the PRB is installed in an appropriate location, could effectively the remediation result[7-9].

Line 62: the phrase “In this study, by using…” is too long and not well constructed.

Revise the following sentence in the introduction section:

In this study, EK-PRB with Fe/Mn/C-LDH materials as PRB filler was employed to remediate the arsenic-contaminated soil. The effects of the operation conditions such as electric field strength, PRB position and moisture content were systematically in-vestigated. To reveal the removal mechanism arsenic from contaminated soil by EK-PRB, the Fe/Mn/C-LDH after EK and reaction was characterized by various tech-niques including scanning electron microscope, x-ray diffractometer, fourier transform infrared, zeta potential analyzer.

Line 64: The phrase “After reaction…” is also too long and poorly written.

Revise the following sentence in the introduction section:

To reveal the removal mechanism arsenic from contaminated soil by EK-PRB, the Fe/Mn/C-LDH after EK and reaction was characterized by various tech-niques including scanning electron microscope, x-ray diffractometer, fourier transform infrared, zeta potential analyzer.

The authors are providing the core information only at the end of phrase, which should be the opposite.

Revise the following sentence in the introduction section:

In this study, EK-PRB with Fe/Mn/C-LDH materials as PRB filler was employed to remediate the arsenic-contaminated soil. The effects of the operation conditions such as electric field strength, PRB position and moisture content were systematically in-vestigated. To reveal the removal mechanism arsenic from contaminated soil by EK-PRB, the Fe/Mn/C-LDH after EK and reaction was characterized by various tech-niques including scanning electron microscope, x-ray diffractometer, fourier transform infrared, zeta potential analyzer.

Line 71: “taken from” is too informal. The authors should be using more proper scientific writing.

Change the expression:

“taken from” was change to “ collected from”

Line 73: why sampling from 5-20 cm? what is the reason to remove the 0-5 cm soil? And why sampling this depth (any particular reason)? Please specific.

Explanation:

Many thanks for the nice question.

The purpose of such sampling is to avoid the influence of decaying animal and plant corpses, sand and gravel on soil sampling, and to ensure the uniformity of sampling. Thus, the effects of moisture content, PRB position and voltage on EK-PRB can be studied more accurately.

Line 70 to 78: the authors must provide more information about the soil characteristics, such as the complete soil classification (by either the US soil taxonomy or the WRB system). Also provide the geology, geomorphology, and climate characteristics of the field site sampling.

Add the follow sentence in the SOIL Section:

Guangxi Zhuang Autonomous Region. The sampling flied site belongs to the Karst geomorphology area, located in the Huanjiang mountain area, belongs to the subtropical monsoon climate area, and has abundant rainfall. The soil samples were taken at a sampling depth of 5-20 cm, and dried naturally after removal of stones and plant residues, named latosol, which is classified as Orthox, Kandiudults and Kanhapludults by the US soil taxonomy.

The methods topics 2.1 to 2.3 must go through a detailed English and style correction and better articulating of ideas. The topics seems to be launched as be bullet points in the text.

Proofread the paper.

We have consulted experts in English for checking the typo, grammar, syntax and sentence structure of our manuscript.

Line 145: the SEM and EDS are very good.

Many thanks for the confirmation from anonymous reviewers.

Line 209: Starting a phrase with “Fig.6(A) illustrated” is not a good way of writing science.

“Fig.6(A) illustrated…”was changed as “The leaching toxicity concentration and removal rate after EK-PRB remediation were illustrated in Fig.6(A) and Table S4.”

Line 218: similar observation as above for starting the phrase with “Fig.6(B) compared” Please make sure to improve this structure in other parts of the paper (lines 230, 241, 247, and others)

The similar writing sciences were improved.

From the topic 3.2 and beyond (for the results and discussion) the reading becomes very tiring. The authors did not make a good job in articulating the ideas.

Proofread and rewrite the text part from topic 3.2 in the manuscript.

The figures 6, 7, 8, 9 are very similar and confusing. Better diagrams must be done to better present the data.

Proofread, rewrite the text part of 3. results and discussion. Redraw the figures.

Line 411: that is not a nice way to start the conclusions with “In this study…”

The conclusions are too short and seems to be quite similar to what is presented in the abstracted (maybe copied?)

Rewrite the conclusion.

References: 36 references are a very small number for a paper. The authors cited only 1 paper from 2021 and none from 2022. More of the state of the art in the last few years and months must be in the paper.

Four more references the last few years were stated.

[35] Nasiri A, Jamshidi-Zanjani A, Darban A K. Application of enhanced electrokinetic approach to remediate Crcontaminated soil: Effect of chelating agents and permeable reactive barrier. Environmental Pollution, 2020, 266: 115197.

[38] Xie N, Chen Z, Huang H, et al. Activated carbon coupled with citric acid in enhancing the remediation of Pb-Contaminated soil by electrokinetic method. Journal of Cleaner Production, 2021, 308: 127433.

[39] Yu Q, Li H, Zheng Y, et al. Promoted electrokinetic treatment of Cr from chromite ore processing residue with rhamnolipid: Focusing on the reactions on electrolyte-residue interfaces. Journal of Environmental Chemical Engineering, 2022, 10: 106954.

[40] Gao M, Zeng F, Tang F, et al. An increasing Cr recovery from soil with catholyte-enhanced electrokinetic remediation: Effects on voltage redistribution throughout soil sections. Separation and Purification Technology, 2020, 253: 117553.

Add the follows sentences in the 3.3 Mechanism analysis part:

And exchangeable and carbonate fractions of As[35]

a high pH was observed near the cathode causing the metal precipitation near the cathode, thereby blocking soil pores and hindering the remediation process[38].

electric current had a dominant effect on temperature changes, and the moisture content rose with the reduction of pH[39],

he Arsenic distribution was concordant with the voltage drops distribution, which indicated the voltage loss took a main role in Arsenic migration[40].

Round 2

Reviewer 1 Report

This is a revsied manscript. I have read it carefully agian, and found that its English writing has been improved significantly.  I hope they will check the whole manuscript once more and improve it further before being accepted by the journal. 

Line 81: flied site??

Line 85-86: Orthox? There is not such suborder of soils in Soil Taxonomy of USDA. All the classification of soils in the study area should be checked by the Keys to the Soil Taxonomy by USDA.

Line 94: Should be "Then".

Line 96: Should be "Afterwards"

Line 98: Should be “, and the solution pH--- ”

------

Author Response

Reviewers' Specific comment

Our response and revision

(Revise according to reviewers’ advices)

Line 81: flied site??

Deleted the incorrect word.

Delete the word of “filed”.

Line 85-86: Orthox? There is not such suborder of soils in Soil Taxonomy of USDA. All the classification of soils in the study area should be checked by the Keys to the Soil Taxonomy by USDA.

Revise the following sentence in the text:

named latosol, which is classified as kandiudults of udults by the US soil taxonomy.

Line 94: Should be "Then".

Revise the word in the text:

Then, the bamboo charcoals were soaked in concentrated

Line 96: Should be "Afterwards"

Revise the word in the text:

Afterwards, the bamboo charcoal bio-templates were immersed in deionized water

Line 98: Should be “, and the solution pH--- ”

Revise the word in the text:

a mixed solution of Fe and Mn(Fe/Mn molar ratio = 2:1) with total concentration of 1 M was added. subsequently, and the solution pH was adjusted to 11.5 by the mixture solution

Reviewer 2 Report

Referree 's suggestion were taken into account.

Author Response

We appreciate the reviewer very much for the comment.